# A Compact and Wideband Active Asymmetric Transmit Array Unit Cell for Millimeter-Wave Applications

**DOI:** 10.3390/s25165168

**Published:** 2025-08-20

**Authors:** Fahad Ahmed, Noureddine Melouki, Peyman PourMohammadi, Hassan Naseri, Tayeb A. Denidni

**Affiliations:** Institut National de la Recherche Scientifique (INRS), Montreal, QC H5A 1K6, Canada; ahmed.fahad@inrs.ca (F.A.); noureddine.melouki@inrs.ca (N.M.); peyman.pourmohammadi@inrs.ca (P.P.); hassan.nasseri118@gmail.com (H.N.)

**Keywords:** transmit array 1, active unit cell 2, beamforming 3

## Abstract

This study presents a compact reconfigurable asymmetric unit cell designed for millimeter-wave (mm-wave) transmit array (TA) antennas. Despite its compact size, the proposed unit cell achieves a broad bandwidth and low insertion loss. By breaking the symmetry of the unit cell and by implementing two MA4AGP910 pin diodes in the proposed unit cell, a phase difference of 180 degrees (1-bit configuration) is obtained in a wide frequency band. The unit cell is fabricated using an LPKF laser machine and characterized using WR-34 waveguide. Measurement results closely match those obtained by simulations, confirming the design’s accuracy. With these functionalities, the proposed 1-bit unit cell emerges as a promising candidate for mm-wave transmit array antennas.

## 1. Introduction

In recent years, with an increasing number of users and smart devices, high-gain and pattern-reconfigurable antennas have received a lot of attention because of their ability to direct and scan the signal in a specific direction [1,2,3]. By incorporating active components, such as PIN diodes, MEMS, and varactor diodes, it is possible to control the radiation null toward the noise source, which allows changing the direction of the main beam and suppressing unwanted signals. Through this, the communication quality and signal-to-noise ratio (SNR) in wireless systems can be significantly improved.

In this perspective, to achieve radiation pattern diversity and enhance the main beam directivity, the reflect array (RA) and transmit array (TA) antennas with different feeds [4,5,6,7,8,9,10,11,12] have been proposed. Similarly, various planar RAs and TAs based on frequency selective surface (FSS) and metasurface with 1-bit [13,14,15], 2-bit [16,17,18,19], and 3-bit phase quantization [20] phase profiles have also been reported in the literature for flexible beamforming and tilting. Higher-bit reconfigurable antenna arrays, such as 3-bit and 4-bit, may offer better performance; however, they also nearly triple or quadruple the cost and system complexity due to the increased number of biasing lines and longer production time.

In contrast, 1-bit and 2-bit reconfigurable arrays can offer comparable performance with reduced system cost and complexity. Therefore, various designs have been reported in the literature offering beamforming functionality using 1-bit or 2-bit configurations [21,22,23,24]. Nguyen et al. have proposed a twistable polarization wideband 1-bit unit cell design offering low insertion loss in the X-band [21]. Another 2-bit reconfigurable unit cell has been designed using the phase control method, which offers a 3 dB bandwidth around 13.9% [22]. Rana et al. have reported an experimental characterization of 2 × 2-unit cells for the X-band applications offering binary phase and insertion loss of -1.87 dB [23]. Similarly, a transmit unit cell employing liquid metal has been presented in the frequency band of 3.2 to 3.43 GHz, offering low insertion loss [24]. Most of the designs suffer from a narrow bandwidth, larger size, or low performance.

On the other hand, utilizing millimeter-wave (mmWave) frequencies can improve the bandwidth and performance of the communication systems [25,26]. The mmWave frequencies provide a wider available spectrum and compact radio frequency (RF) transceiver designs. To further address the bandwidth problem, frequency selective surface (FSS) or metasurface-based transmitting antenna arrays have been studied and reported in the literature. Thus, various 1-bit and 2-bit unit cells have been proposed in the K-band.

A 1-bit reconfigurable transmissive unit cell for Ka-band applications has been reported in [27]. The unit cell size at the center frequency is 0.49 × 0.49 λ^2^ (λ is free space wavelength), and its performance is acceptable only within the frequency band of 27.7 to 29.7 GHz, with an 11.1% 3 dB fractional bandwidth. Di Palma et al. [28] have proposed an experimental characterization of a 1-bit unit cell for Ka-band applications. However, its size is compact but 3 dB bandwidth is merely 12%. Similarly, a 2-bit unit cell for a transmit array antenna has been reported, achieving 16.2% 3 dB fractional bandwidth at a cost of four pin diodes and a bulky structure [29]. Another experimental validation of a 2-bit reconfigurable unit cell has been proposed for the Ka-band [30], using four pin diodes and having very high insertion loss. Moreover, the 3 dB bandwidth is merely 10.1%. Recently, a wideband 1-bit unit cell has been reported in the literature [31], achieving wideband operation using multiple layers and four pin diodes, but it only achieves 180° phase difference if the polarization state is twisted, which makes this design unsuitable for designing transmit array antennas.

However, all these designs achieve 1-bit or 2-bit configurations, but most of them have a very low 3 dB bandwidth, larger unit cell size, multilayer structure, and more diodes, which not only affects the array performance, but also increases the overall cost of the communication system. It is also pertinent to mention here that all the above designs have high insertion loss, and even with the 3 dB criterion, their bandwidth is low. At 3 dB almost half of the wave is reflected, which not only affects the source feed, but also decreases the overall performance of the transmitting array. Moreover, 2-bit and 3-bit designs suffer from a higher insertion loss due to the requirement of a greater number of diodes. As is obvious from this survey paper [32], a 1-bit phase quantization unit cell can be generally implemented in practical state-of-the-art electronically reconfigurable transmit arrays. This is because 1-bit quantization can be realized using a minimal number of switches and metal layers, making it a practical solution in terms of implementation complexity and cost. Therefore, to design a 1-bit transmissive unit cell with a compact size, reduced cost, and low insertion loss at mmWave frequencies is still a challenging topic nowadays.

In this work, we present an asymmetric-architecture-based 1-bit transmit array unit cell at mmWave frequencies that combines current reversal with a simplified two-diode biasing scheme, achieving compact size, wide 2 dB bandwidth, low insertion loss, and consistent full-band 180° phase shift without polarization change. Operating across 25–31 GHz, the design maintains transmission above –2 dB for both diode states. Its reduced footprint not only minimizes the overall size of the transmit array but also helps avoid grating lobes. The performance is first validated through WR-34 waveguide measurements, followed by integration into a complete phase-gradient surface for beamforming analysis using full-wave electromagnetic (EM) simulation, confirming its suitability for high-performance mmWave applications.

## 2. Unit Cell Design and Geometric Configurations

### 2.1. Unit Cell Design

This section presents the geometrical configuration of a linearly polarized reconfigurable unit cell with a 1-bit configuration (0° and 180°) achieved through combining symmetry breaking and current-reversal techniques. The proposed unit cell is composed of two stacked Rogers 5880 dielectric substrates, and four metallic layers, as shown in Figure 1. These two substrates are bonded together using an adhesive RO4450F material with a thickness of 0.101 mm and dielectric constant 3.52, which not only provides mechanical strength, but also maintains proper isolation between the layers. The top substrate has a thickness of 1.575 mm and a dielectric constant (r) of approximately 2.2. As shown in Figure 1a, the top layer contains the active components, MA4AGFCP910 (MACOM) PIN diodes, incorporated in a three top metal strip, which functions as a receiving layer. The bottom substrate is also composed of Rogers 5880 substrate with a reduced thickness of 0.508 mm. As shown in Figure 1b, this layer includes bottom passive rectangular copper patches and is known as the transmitting layer. The top layer active patch is connected to the bottom layer passive patch through a via (diameter of the via is 0.3 mm), which is placed at the center of the metallic part of the unit cell, as illustrated in Figure 1c. A ground plane contains two intermediate layers; one is a complete copper with a 0.4 mm diameter hole, which allows the via between active and passive layers. The second intermediate layer contains biasing lines, as depicted in Figure 1c.

As shown in Figure 1a, the asymmetric dimension of the reconfigurable unit cell is chosen as Px = 5 mm × Py = 4 mm, which permits control over surface impedance matching, thereby enhancing the transmission bandwidth by reducing the reflections [33]. This type of unit cell not only enables a wider bandwidth, but also contributes to compacting the overall size of the transmitting antenna arrays, resulting in improved aperture efficiency. The unit cell transmits the incident wave from the top receive layer to the bottom transmit layer through the via. Switching the ON/OFF configuration of the pin diode reverses the current, resulting in a phase shift from 0° to 180°. The feeding lines are connected to the top layer active patch using a metallic via of diameter 0.2 mm. In the proposed asymmetric unit cell design, a 9.97 mA current is used to forward-bias a one PIN diode, and the 1.2 V threshold voltage effectively disables the reverse-biased diode, which is arranged in an anti-parallel configuration. In large-scale array implementations, this arrangement greatly simplifies the design and provides a biasing network by enabling the use of only one bias line per unit cell [34].

The optimized dimensions of the unit cells are as follows: a = 0.77 mm, b = 1 mm, c = 0.35 mm, d = 0.88 mm, Lf = 3.3 mm, Lbx = 3.1 mm, Lby = 2.5 mm, and s = 0.4 mm. Moreover, the gap between the top layer strips to place the diodes is kept at 0.475 mm. The width of the connection and feeding lines is chosen to be 0.1 mm with a gap of 0.2 mm to allow simple fabrication through LPKF.

### 2.2. Diode Characterization and Circuit Model

The MA4AGP910 pin diode from M/ACOM [35] is selected due to its compact dimensions, high isolation, and low insertion losses. The measurement setup to obtain the insertion loss and isolation of this pin diode is shown in Figure 2a,b. The insertion loss and isolation of the diode at 10 mA and −1.2 V are depicted in Figure 3a,b. Moreover, the equivalent lumped element model of the PIN diode during the ON and OFF states is presented in Figure 4a,b.

In the reverse (V_bias_) and forward (I_bias_) states, the circuit is modeled as a shunt circuit (*Z_OFF_* = [C*t* = 42 fF, R*p* = 300,000 Ω]) and a series circuit (*Z_ON_ =* [L*d* = 0.0499 nH, R*s* = 4.2 Ω]), respectively. The manufacturer’s specification is satisfied by the experimental value of R*s* = 4.2 Ω, which mainly sets the insertion losses [35]. Due to different characterization conditions, the other parameters vary from these specifications. The manufacturer’s specification is satisfied by the experimental value of Rs = 4.2 Ω, which primarily determines the insertion losses [35]. Due to different characterization conditions, the other parameters show slight variation from the specified values. The diode characterization was performed on a Rogers RO5880 substrate (εr = 2.2 and tanδ = 0.0009) with a thickness of 0.508 mm.

### 2.3. Fabrication Prototype and Measurement Enviroment

The main unit cell is used for the full array, but for WR-34 waveguide measurements, slight modifications such as reducing the Py dimension and adding an extra via for feeding are made to minimize leakage and adapt to the setup, as shown in Figure 5a. This configuration keeps the feeding pads on the same side and avoids short circuits in the waveguide; however, these changes led to a small increase in insertion loss of 0.4 dB and a 30 MHz frequency shift, which can be seen from Figure 5b. To verify the unit cell performance in a waveguide, a pair of unit cells is fabricated, as shown in Figure 6a,b, and characterized using a WR-34 waveguide (4.3 × 8.6 mm^2^). To ensure the continuity of the waveguide walls in the fabricated prototype, a row of vias with approximately 2 mm diameter is placed along all sides, and a larger ground plane is implemented, as shown in Figure 6a,b. These vias make contact with the flanges of the WR-34 waveguide to prevent the leakage and propagation of substrate modes outside the unit cells. Due to the relatively large via diameter compared to the edges of the unit cell, current may be disturbed by these vias, allowing potential RF/DC leakage through the feed lines. Various techniques such as meander lines, radial stubs, and resistive lines are evaluated. However, the meander line and radial stubs exhibited poor isolation because of the compact unit cell structure and limited available space. Conversely, a resistive sheet resulted in very high transmission loss. High-impedance biasing lines were employed in the structure along with a commercially available inductor (L02013R3BHSTR), providing effective isolation and low transmission loss while preventing both DC and RF signal leakage.

To test the proposed prototype, Thru-Reflect-Line (TRL) calibration is performed, and then the prototype is placed in the WR-34 waveguide to obtain the measurement results, as illustrated in Figure 7. The vector network analyzer (VNA) is used to transmit and receive the signal, whereas to control the on/off state of the pin diodes, a power supply is used.

## 3. Results and Discussions

### 3.1. Unit Cell Results

The full-wave electromagnetic (EM) simulation has been performed using EM software CST studio 2023 to obtain the scattering parameter results of the proposed unit cell. Two different cases are studied, as shown in Figure 8a,b. Initially, the WR-34 waveguide flanges, along with the outer part of the unit cell, including vias and pads, are considered in the simulation to obtain the transmission (S21) and reflection (S11) results, as shown in Figure 8a. The S21 of this configuration is observed at approximately −0.81 dB, and S11 is less than −10 dB within the desired frequency band of 25 to 32 GHz. Subsequently, the proposed unit cells are incorporated into the complete setup, as illustrated in Figure 6b, to achieve the S-parameters and phase results. It can be seen from Figure 9a,b that S21 remains above −2 dB in the frequency band of 25.5 to 30.3 GHz and the phase difference of 180^0^ is achieved over a wide fequency band. Moreover, the S11 remains below −10 dB in the complete desired band. To achieve the 1-bit configuration (180° phase difference), the states of the PIN diodes are varied between ON and OFF.

To better understand the 1-bit configurations, at central frequency, the surface current distribution of the unit cell is plotted in Figure 10a–d. As obvious from Figure 10a,b, when the diode state is ON/OFF, the current flows from left to right for both the receive and transmit metallic patches. By switching the state of diodes from ON/OFF to OFF/ON, the direction of surface current on both the receive and transmit unit cell patches is completely reversed, as shown in Figure 10c,d, resulting in a 180° phase shift.

The surface current distribution of the unit cell is shown in Figure 8. The transmission coefficient remains above −2 dB for both diode configurations (i.e., ON/OFF and OFF/ON), as depicted in Figure 11a. Additionally, a phase difference of 180° is achieved across the frequency band of 25 to 31 GHz, as shown in Figure 11b.

To practically validate the simulation results, measurements were carried out to evaluate the actual performance of the unit cell in a real environment. The measurement setup is illustrated in Figure 7, comprising two WR-34 waveguide flanges, the proposed unit cell, and connecting wires to control the diode states. The setup was connected to a vector network analyzer (VNA) to observe the response of the unit cell. The transmission coefficient is observed close to −2 dB for both configurations of the diodes (i.e., ON/OFF and OFF/ON), as depicted in Figure 11a. The measured S21 results for both diode configurations fall within the target frequency band of 25 to 31 GHz. However, the 2 dB transmission bandwidth is observed from 25.4 to 30.2 GHz.

Furthermore, the measured phase difference between the two diode configurations is approximately 180° across the entire frequency range of 25 to 31 GHz, which can be seen from Figure 11b. Minor variations in the transmission, reflection, and frequency shift could be attributed to imperfections in diode placement and the measurement setup.

### 3.2. Phase Gradient Surface

To verify the beamforming performance of the proposed asymmetric unit cell, an array of 10 × 12 elements is considered, as shown in Figure 12a,b. The feeding lines and all the switches are considered so that all types of losses could be included in the simulation. The schematic of the proposed simulation setup is illustrated in Figure 13. A transmitting aperture comprising an M × M planar array and a source is placed at $z={\vec{r}}_{f}$. The proposed unit cell is considered as an active phase shifter to control the beam direction emitting from the source. According to the far-field theory presented in [36], the far field pattern of the antenna can be represented as(1)$$F\left(ɵ,\Phi \right)=\sum_{m=1}^{M}\sum_{n=1}^{M}\frac{f_{e}(\theta ,\varphi )f_{f}(\theta ,\varphi )\left|T_{m,n}\right|}{\left|{\vec{r}}_{m,n\;}-{\vec{r}}_{f}\right|}\times exp\;(jk\left|{\vec{r}}_{m,n\;}-{\vec{r}}_{f}\right|-{\vec{r}}_{m,n\;}\cdot \hat{u}+j\varphi_{m,n})$$
where $f_{f}\left(\theta ,\varphi \right)$ is the radiation source, $f_{e}(\theta ,\varphi )$ denotes the active unit cell and $\hat{u}=\hat{x}sin$θcos*φ* + $\hat{y}sin$θsin*φ* + $\hat{z}cos$θ. The vectors ${\vec{r}}_{f}$ and ${\vec{r}}_{m,n}$ represent the position of the source and the mn-th unit cells, respectively. $\left|T_{m,n}\right|$ is the transmission of the unit cell and k is the free pace wave constant.

To direct and control the beam at the desired angle, the phase of the individual element should be adjusted, as obvious from Equation (2).(2)$$\varphi_{m,n}=k(\left|{\vec{r}}_{m,n\;}-{\vec{r}}_{f}\right|-{\vec{r}}_{m,n\;}\cdot \hat{u})+\varphi_{c}$$
where $\varphi_{c}$ depicts the phase constant, which should be the optimized value of the 1-bit, 2-bit or 3-bit transmit array antennas, which further depend on the relative phase of the transmission instead of an absolute one.

The phase distributions used to tilt the beam in the desired directions are obtained using a MATLAB R2023a code and are plotted in Figure 14a–c. The simulations are performed considering the complete array and a source placed at a focal distance of F = 57. The radiation patterns corresponding to different phase distributions are shown in Figure 15a.

The black curve represents the beam focused at 0°, while the blue curve denotes the beam tilted at 15°. Similarly, the red and green curves represent beam tilts at 30° and 45°, respectively. As shown in Figure 15b, the peak gain at the central frequency is observed around 23 dB and the side lobe levels remain low.

Table 1 provides a comprehensive comparison of the proposed transmit unit cell against state-of-the-art transmit array unit cells. The comparison is based on the 2 dB bandwidth, overall size, and thickness of the unit cell, and the number of diodes. It is clear from Table 1 that the proposed unit cell has a wider 2 dB bandwidth compared to already published designs. Moreover, the size of the unit cell is compact, which not only avoids the grating lobes, but also reduces the array footprint. Moreover, using fewer unit cells reduces the overall cost of the design.

## 4. Conclusions

In this paper, a compact, reconfigurable 1-bit transmit unit cell with an asymmetric structure has been presented for millimeter-wave (mmWave) transmit array (TA) antennas. Despite its small size, the unit cell has achieved a broad bandwidth and low insertion loss across the 25–31 GHz frequency range, helping to reduce the overall array size and prevent grating lobes. By breaking symmetry and incorporating two MA4AGP907 pin diodes, a 180° phase difference (1-bit configuration) has been realized over a wide frequency band. The unit cell has been fabricated using an LPKF laser machine and characterized within a WR-34 waveguide, with measurement results closely matching simulations and validating the design’s accuracy. Integrated into a complete phase-gradient surface, beamforming analysis using electromagnetic simulation confirms its performance. These combined features make the proposed design a strong candidate for mmWave transmit array applications.

## Figures and Tables

**Figure 1 sensors-25-05168-f001:**
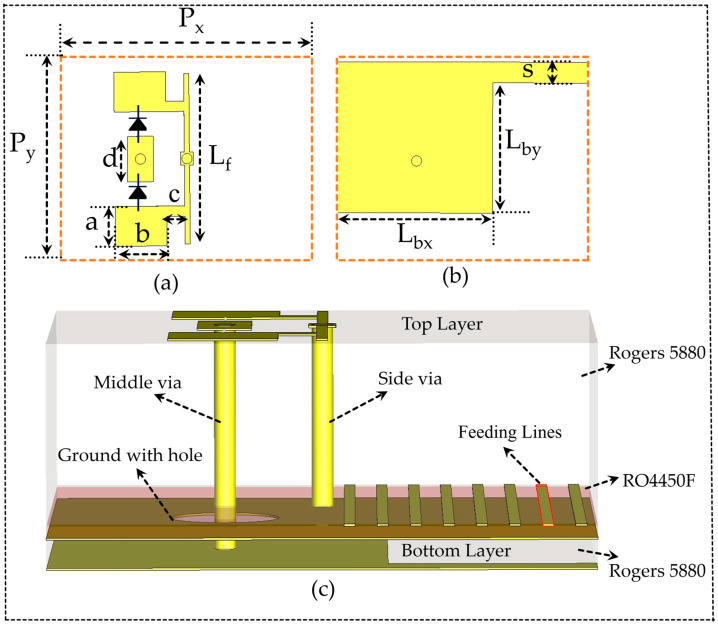
(**a**) Front view of the proposed active unit cell; (**b**) backside view of the proposed active unit cell; (**c**) side view of the proposed unit cell.

**Figure 2 sensors-25-05168-f002:**
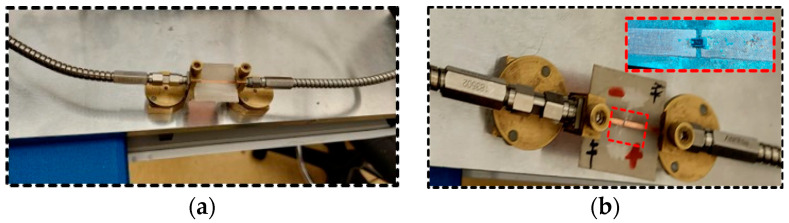
(**a**) Co-planar transmission line; (**b**) diode characterization setup.

**Figure 3 sensors-25-05168-f003:**
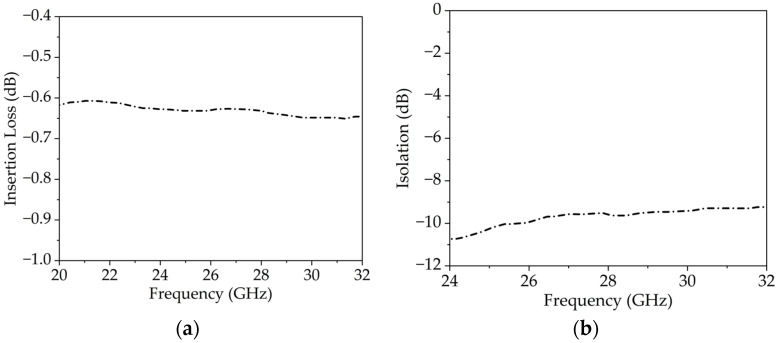
(**a**) Insertion loss of the PIN diode at 10 mA bias current, and (**b**) isolation of the PIN diode at –1.2 V bias voltage.

**Figure 4 sensors-25-05168-f004:**
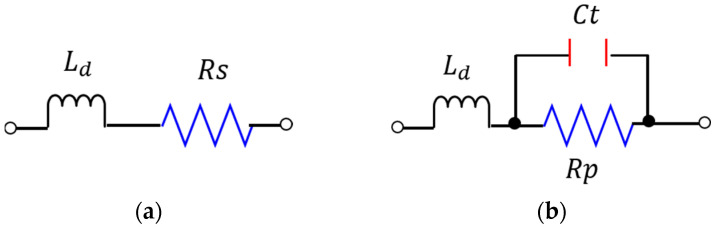
Equivalent model of PIN diode: (**a**) forward bias (*Z_ON_*); (**b**) reverse bias (*Zoff*).

**Figure 5 sensors-25-05168-f005:**
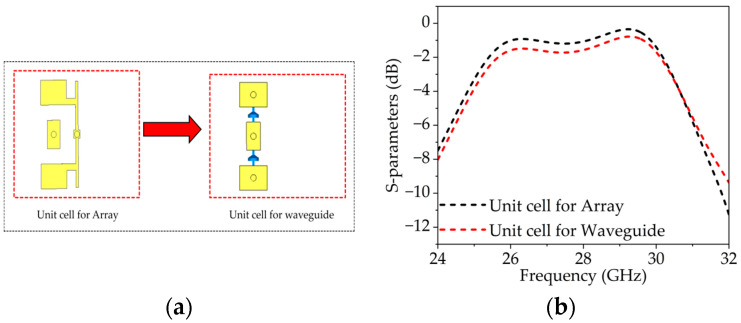
(**a**) Changes in biasing configuration with respect to WR-34 waveguide. (**b**) Transmission coefficient for both the unit cells.

**Figure 6 sensors-25-05168-f006:**
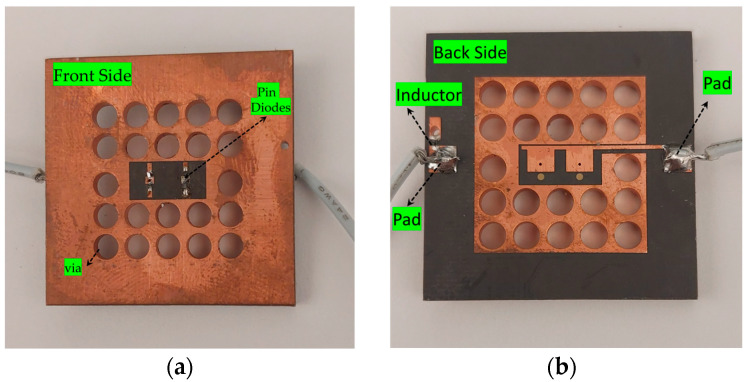
(**a**) Front side of the proposed unit cell; (**b**) back side of the unit cell.

**Figure 7 sensors-25-05168-f007:**
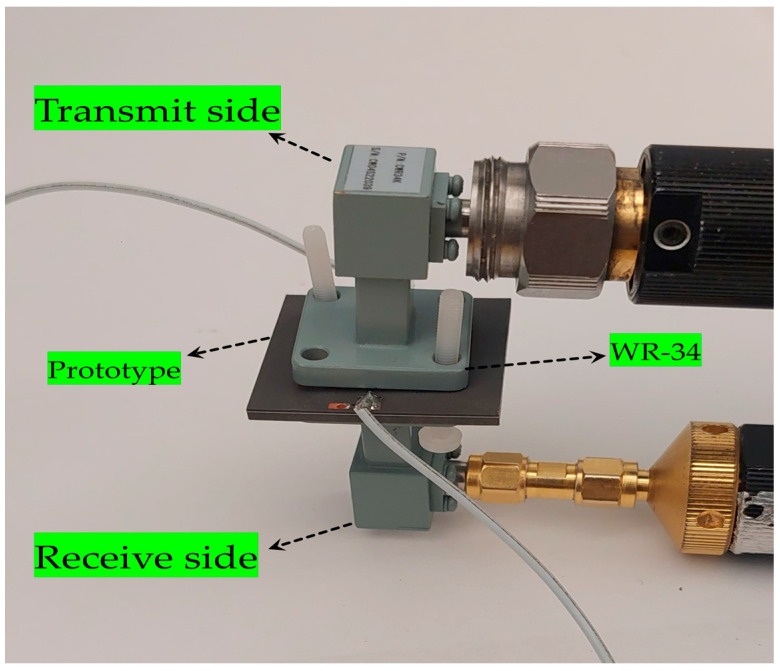
Measurement setup of the asymmetric unit cell.

**Figure 8 sensors-25-05168-f008:**
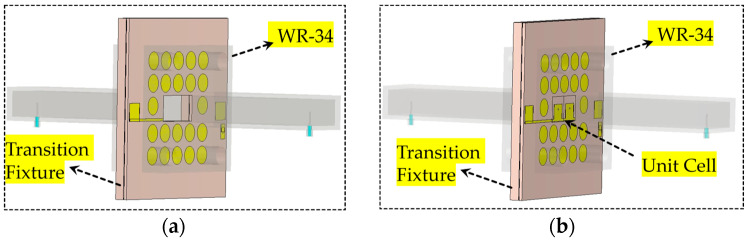
(**a**) Transition fixture without unit cells; (**b**) transition fixture with unit cells.

**Figure 9 sensors-25-05168-f009:**
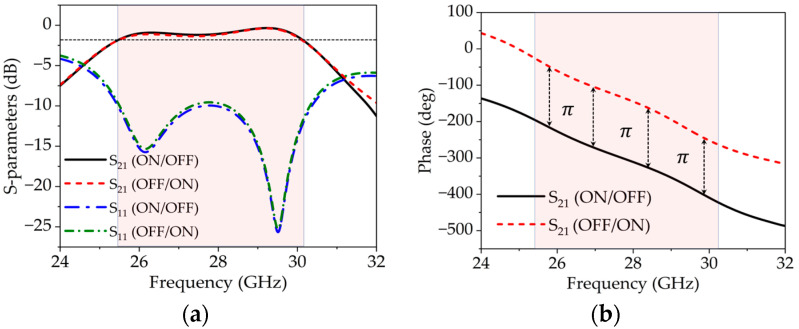
(**a**) Scattering (S)-parameters of the proposed unit cell in the WR-34 waveguide; (**b**) phase of the transmission for both the ON/FF and OFF/ON states.

**Figure 10 sensors-25-05168-f010:**
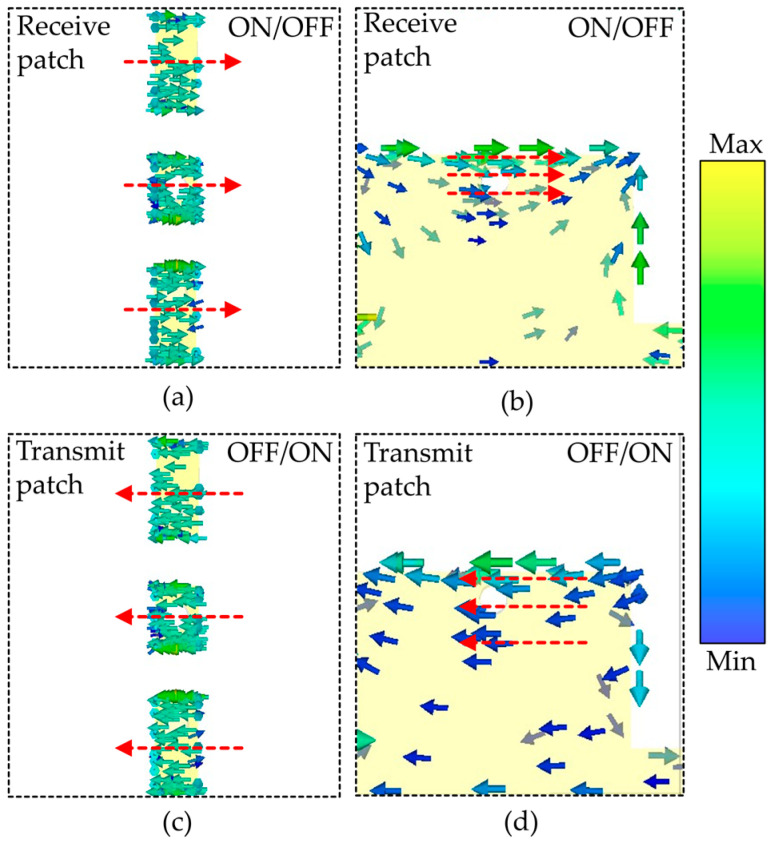
(**a**) Current distribution on receive layer when diodes are ON/OFF; (**b**) current distribution on transmit layer when diodes are ON/OFF; (**c**) current distribution on receive layer when diodes are OFF/ON; (**d**) current distribution on transmit layer when diodes are OFF/ON.

**Figure 11 sensors-25-05168-f011:**
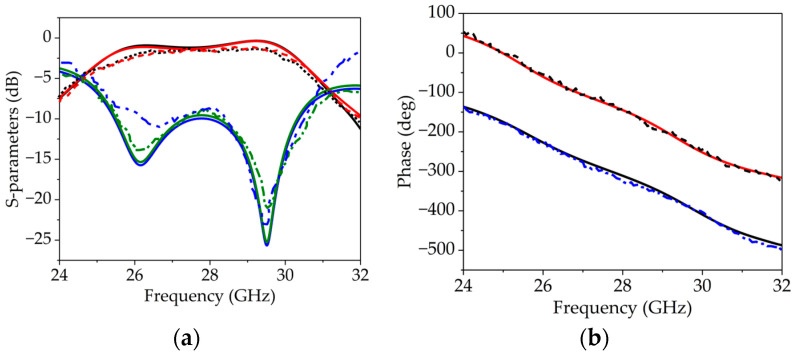
(**a**) Simulation and measurement results of s-parameters. (**b**) Simulation and measurement results of different phases. (Note: The solid curves are simulated results and dotted curves are measurement results.).

**Figure 12 sensors-25-05168-f012:**
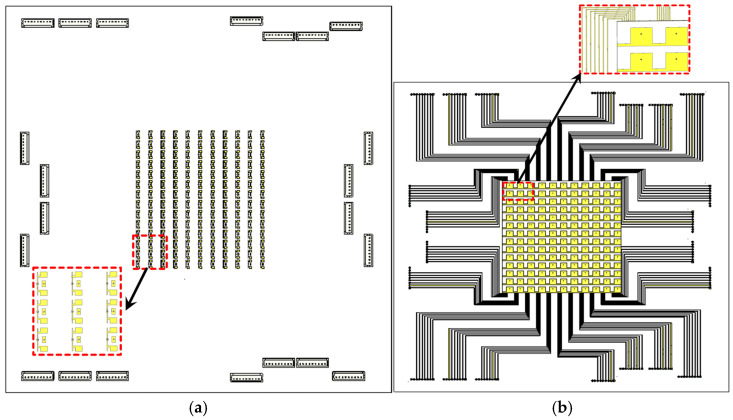
(**a**) Receiving side of the proposed array; (**b**) transmitting side of the proposed array.

**Figure 13 sensors-25-05168-f013:**
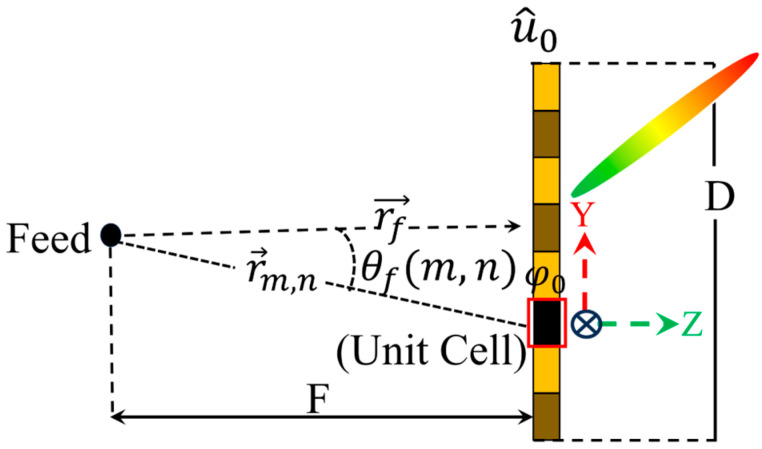
Schematic diagram of the simulation setup.

**Figure 14 sensors-25-05168-f014:**
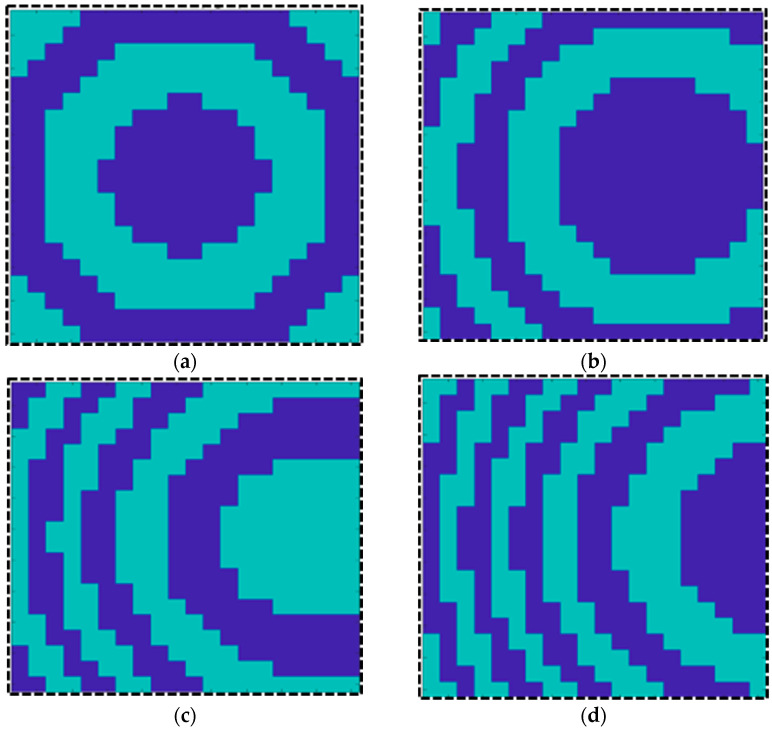
(**a**) Coding sequence for 0 degrees; (**b**) coding sequence for 15 degrees; (**c**) coding sequence for 30 degrees; (**d**) coding sequence for 45 degrees.

**Figure 15 sensors-25-05168-f015:**
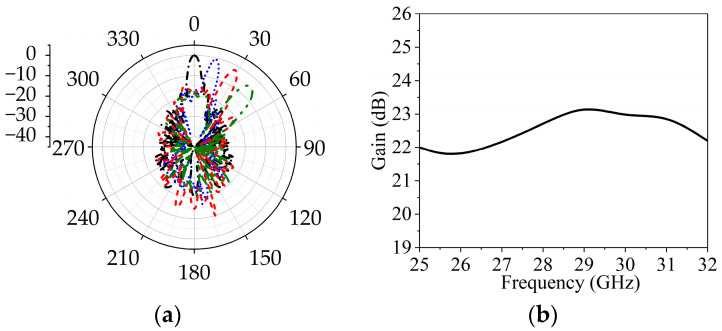
(**a**) Beam scanning at different angles at central frequency; (**b**) peak gain of the proposed array.

**Table 1 sensors-25-05168-t001:** Comparison with previously published designs.

Reference	No of Diodes	Max Gain (dB)	AE	BS Range	1 dB BW	2 dB BW	3 dB BW	Element Thickness	Element Size
[21]	2	N/A	N/A	N/A	5%	<14%	26%	0.16 λ	0.65 λ × 0.65 λ
[22]	4	20	36%	±45	2%	<9%	13%	0.21 λ	0.533 λ × 0.533 λ
[23]	2	N/A	N/A	N/A	1.25%	<5%	7%	0.124 λ	0.52 λ × 0.52 λ
[27]	2	N/A	N/A	N/A	1%	<7%	11%	0.15 λ	0.527 λ × 0.527 λ
[28]	2	N/A	N/A	N/A	6.5%	<8.5%	13.8%	0.151 λ	0.51 λ × 0.51 λ
[29]	4	19.8	15.9%	±60	3%	<8.9%	16.2%	0.21 λ	0.51 λ × 0.51 λ
[30]	4	N/A	N/A	N/A	0%	<4%	10.1%	0.21 λ	0.52 λ × 0.52 λ
[31]	4	N/A	N/A	N/A	0%	<17%	30%	0.249 λ	0.37 λ × 0.37 λ
[37]	2	18	12.7%	±60	5%	<9%	15.22%	0.06 λ	0.247 λ × 0.247 λ
This work	2	23	40.2%	±45	7%	18.24%	21.43%	0.19 λ	0.37 λ × 0.47 λ

Here λ is free-space wavelength; BW: bandwidth; BS: beam steering; AE: aperture efficiency.

## Data Availability

No data available.

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
