# Peer review of "A Compact and Wideband Active Asymmetric Transmit Array Unit Cell for Millimeter-Wave Applications"

_sensors, 2025, doi:10.3390/s25165168_

Round 1

Reviewer 1 Report

Comments and Suggestions for Authors

This paper presents the design and analysis of a reconfigurable asymmetric unit cell for transmit array antennas at millimeter-wave frequencies with size compactness. Please find my comments as follows:

  1. In section 2.2, the authors have discussed the equivalent circuit model and corresponding parameters at on- and off-states. It is required that the authors shall include the benchmark for model extraction and include the figure representing the equivalent circuit model.
  2. The experimental results have demonstrated a nice performance with the reconfiguration using commercial GaAs PIN diode. However, an obvious degradation at the lower frequency seems to be quite extreme. Since the diode was available for operation up to 67 GHz, please discuss the major reason resulting in this behavior.
  3. The originally designed operating frequency ranges from 26-30 GHz, but the peak gain for the proposed array seems to be falling at 29-31 GHz. Please explain why the frequency has shifted up to higher operating frequencies.
  4. Please also include the comparison of 1- and 3-dB bandwidth in the comparison table. The experimental results demonstrate a nice performance compared to the state-of-the-art works. It would be nice to know how the proposed design behaves when considering different bandwidth requirements.

Author Response

Response Letter attached to it. 

Reviewer 2 Report

Comments and Suggestions for Authors

This manuscript propose a compact 1-bit reconfigurable transmit array unit based on asymmetric structures and current reversal technology. The design operates in the frequency range of 25 GHz to 31 GHz with low insertion loss and achieves a 180° phase shift. The concept has been validated through simulation and measurements. The advantages of this work lie in its compactness and wide bandwidth. However, due to the following issues, this paper requires major revisions:

  1. The authors claim the proposed unit cell exhibits novelty. However, similar concepts based on asymmetry, current reversal, and 1-bit switching have been reported in the literature [21], [22], [27], [29], and [31]. So, what is the key innovation of the design in this paper? Are the improvements primarily in bandwidth, size, or insertion loss? Please provide quantitative evidence to support your claims. Furthermore, a clearer discussion of the novelty and specific contributions of the design is strongly recommended.
  2. This paper primarily uses electromagnetic simulation and qualitative surface current observations to demonstrate the 180° phase shift. Could the author provide a simplified transmission line or equivalent circuit model to explain the switching mechanism?
  3. The paper proposes the use of MA4AGP910 PIN diodes, however, it is unclear whether the parasitic effects—such as the equivalent inductance and resistance of the diodes were considered in the simulations. Furthermore, the bias circuits of the switching elements are not discussed. These aspects are particularly critical for switch control circuits operating in the millimeter-wave band, where parasitic effects can significantly impact performance. How can consistency between simulation and test results be ensured?
  4. Table 1 The comparison results are not convincing enough. It is recommended to add more comparison indicators, such as actual gain, aperture efficiency, and beam steering range. Also, the number of Table 2 is incorrect.
  5. In both the Introduction and the tables, the symbol λ should be explicitly defined as either the free-space wavelength or the guided wavelength to avoid ambiguity.
  6. There are some grammatical and stylistic issues throughout the manuscript. like “This configuration keep the feeding pads…” should be “This configuration keeps the feeding pads...”; “provides a compact the array size” should be “reduce the array footprint”.

Author Response

Response Letter attached to it.

Round 2

Reviewer 1 Report

Comments and Suggestions for Authors

The reviewer thanks for the efforts that the authors have spent on revising the manuscript based on the comments. The paper can be accepted in its current form.

Author Response

Thank you so much for accepting our paper.

Reviewer 2 Report

Comments and Suggestions for Authors

Although the author has provided responses and revisions, there are still some issues that need to be addressed:

  1. The abstract refers to “MA4AGP907,” while Section 2.1 of the main text refers to “MA4AGFCP910,” and the section on diode characterization and equivalent circuits refers to “” Please carefully review the text.
  2. The paper states, “first implementation of an asymmetric-architecture-based 1-bit TA unit cell at mmWave... (implementation),” but the previous review also acknowledges that related ideas have already been studied. We suggest changing “first implementation” to something like “to the best of our knowledge, among... we achieve...” to make it verifiable, quantifiable, and restrictive.
  3. The author provided the equivalent model configuration and parameter symbols in the reply, but the main text currently only shows diodes (ZON/ZOFF) and brief textual descriptions, which are all available in the manual. Could you provide detailed simulation settings? How can losses be minimized? I am still concerned about ensuring consistency between bias circuit simulation and testing, as parasitic effects and losses are particularly pronounced in the millimeter-wave band. Providing this level of detail would be highly valuable.
  4. There are also many typos. “-1.2” in the title of Figure 3; the text “respectivelyThe manufacturer's ...” in line 149. In addition, please describe the substrate material clearly.

Author Response

Please find the response letter attached to it.
